# Association of Virulence, Biofilm, and Antimicrobial Resistance Genes with Specific Clonal Complex Types of *Listeria monocytogenes*

**DOI:** 10.3390/microorganisms11061603

**Published:** 2023-06-17

**Authors:** Peter Myintzaw, Vincenzo Pennone, Olivia McAuliffe, Máire Begley, Michael Callanan

**Affiliations:** 1Department of Biological Sciences, Munster Technological University, Bishopstown, T12 P928 Cork, Ireland; peter.myintzaw@teagasc.ie (P.M.);; 2Teagasc Food Research Centre, Moorepark, Fermoy, Co., P61 C996 Cork, Ireland

**Keywords:** *Listeria*, stress tolerance, comparative genomic, typing, virulence profile

## Abstract

Precise classification of foodborne pathogen *Listeria monocytogenes* is a necessity in efficient foodborne disease surveillance, outbreak detection, and source tracking throughout the food chain. In this study, a total of 150 *L. monocytogenes* isolates from various food products, food processing environments, and clinical sources were investigated for variations in virulence, biofilm formation, and the presence of antimicrobial resistance genes based on their Whole-Genome Sequences. Clonal complex (CC) determination based on Multi-Locus Sequence Typing (MLST) revealed twenty-eight CC-types including eight isolates representing novel CC-types. The eight isolates comprising the novel CC-types share the majority of the known (cold and acid) stress tolerance genes and are all genetic lineage II, serogroup 1/2a-3a. Pan-genome-wide association analysis by Scoary using Fisher’s exact test identified eleven genes specifically associated with clinical isolates. Screening for the presence of antimicrobial and virulence genes using the ABRicate tool uncovered variations in the presence of *Listeria* Pathogenicity Islands (LIPIs) and other known virulence genes. Specifically, the distributions of *actA*, *ecbA*, *inlF*, *inlJ*, *lapB*, LIPI-3, and *vip* genes across isolates were found to be significantly CC-dependent while the presence of *ami*, *inlF*, *inlJ*, *and* LIPI-3 was associated with clinical isolates specifically. In addition, Roary-derived phylogenetic grouping based on Antimicrobial-Resistant Genes (AMRs) revealed that the thiol transferase (*FosX*) gene was present in all lineage I isolates, and the presence of the lincomycin resistance ABC-F-type ribosomal protection protein (*lmo0919_fam*) was also genetic-lineage-dependent. More importantly, the genes found to be specific to CC-type were consistent when a validation analysis was performed with fully assembled, high-quality complete *L. monocytogenes* genome sequences (*n* = 247) extracted from the National Centre for Biotechnology Information (NCBI) microbial genomes database. This work highlights the usefulness of MLST-based CC typing using the Whole-Genome Sequence as a tool in classifying isolates.

## 1. Introduction

*L. monocytogenes* is an opportunistic, Gram-positive, food-borne pathogen that causes listeriosis, which is a major concern from a food safety and public health perspective. Elderly and immunocompromised people may have greater hospitalisation with sepsis, meningitis, and meningoencephalitis as well as higher fatality rates (13% in 2020) [1] due to the disease. The pathogen can cause symptoms such as fever, muscle aches, and gastrointestinal symptoms such as nausea and diarrhoea. While severe disease is rare, in pregnant women, it can cause miscarriage, stillbirth, preterm labour, sepsis, or meningitis in new-borns. *L. monocytogenes* infections are caused mainly through the consumption of ready-to-eat foods, such as dairy products, processed meat, fish, and fresh produce. This problem is compounded by the microbe’s propensity to survive/grow in stressful conditions (acidic environments, high salt concentrations, and low temperatures), which are commonly used in the food industry to control microbial safety.

The capability of *L. monocytogenes* to proliferate at lower temperatures [2,3] and lower pH [4,5], the resistance to commonly used disinfectants [6] in the food industry, and the ability to form biofilms [7] is primarily responsible for their stress survival and persistence. One of the key problems in microbial risk profiling is strain variation. To distinguish between strains, multiple typing approaches such as serotyping, Multi-Locus Sequence Typing (MLST), and various fragment-based typing methods have been employed. Subtyping allows researchers and public health officials to differentiate between different strains of *L. monocytogenes*, which can help with identifying the source of an outbreak and implementing appropriate control measures. In addition, studies of the genetic diversity of *L. monocytogenes* have helped shed light on key aspects of virulence, persistence, and environmental stress adaptation.

Traditional molecular typing methods such as serotyping and phage typing are labour-intensive, time-consuming, and can produce inconsistent results between methods [5]. They also have a lower resolution compared to genomic analysis with next-generation sequencing (NGS) technology [8,9,10]. MLST based on whole-genome sequences (WGSs) is robust, convenient, and efficient [11] and has been utilised in epidemiological research [12], pathogen surveillance, and food-related stress tolerance investigations [5,13]. In addition, high-throughput Real-Time PCR methods have recently been developed that allow for the rapid determination of *L. monocytogenes* isolate CC-type [14]. Increasingly accessible pan-genome annotation software such as Prokka v1.13.4 [15] and Prokaryotic Genome Annotation Pipeline (PGAP) [16] makes it possible to efficiently and consistently characterise the specific genotype. Furthermore, pan-genome comparative genomic studies offer more insights on phylogeny, virulence, and other differentiating characteristics to help with the better management of this foodborne illness [17,18,19].

Undissociated acid [5], salt, and pH tolerance [20] have already been described for the isolates in this study. However, the differences in genetic makeup related to the critical characteristics of virulence, biofilm formation, and antimicrobial resistance had not been reported with this set of isolates. The aim of this work was to perform comparative genomic analysis of the 150 *L. monocytogenes* isolates from food, food processing environments, and clinical sources to determine specific genetic markers associated with virulence markers, their ability to form biofilms, and antimicrobial resistance based on WGS data. More importantly, associations between CC-type and genetic markers were validated using additional genomes extracted from NCBI.

## 2. Materials and Methods

### 2.1. Strains

A total of 150 *L. monocytogenes* isolates from various food production and clinical sources (Figure 1) were sourced from the *Listeria* collection at Teagasc Food Research Centre, Moorepark, Co Cork. All isolates used in this study were previously published by Myintzaw et al. in 2022 [5]. In addition, reference strains of *L. monocytogenes*, EGDe, 10403S [21], 6179 [22], and F2365 [23] were included. 

### 2.2. Whole-Genome Sequences

For each isolate (*n* = 150), raw reads in fastq format obtained from paired-end libraries and Illumina MiSeq sequencing were available from a previous study [5]. The raw reads were quality-checked and the adaptors were removed using the Trimmomatic tool [24] available on the online platform Galaxy (https://usegalaxy.org/, (accessed on 21 March 2021)) [25]. Genome assemblies were performed using SPAdes [26], also available on the Galaxy online platform (https://usegalaxy.org/, (accessed on 29 March 2021)). 

### 2.3. Pan-Genome Analysis

Prokka was employed to annotate all of the genomes in the UseGalaxy.org online platform using default parameters, and the resultant gff3 files were used for pan-genome analysis with Roary 3.13.0 version [27], available at (http://sanger-pathogens.github.io/Roary/, (accessed on 24 October 2021)). The presence or absence of the pan-genome gene file generated by Roary was compared with the source of isolates in Scoary (Gene-wise counting and Fisher’s exact tests for trait) to establish gene clusters associated with each trait [28], available at (https://github.com/AdmiralenOla/Scoary, (accessed on 21 January 2022)). 

### 2.4. Genotypic Characterisation

Tools available on the Centre for Genomic Epidemiology’s (CGE) online platform (https://cge.cbs.dtu.dk/services/, accessed on 12 December 2022) were used to perform the genotypic characterisation of the isolates utilising core genome cgMLST (cgMLSTFinder) typing based on single-nucleotide polymorphisms at 1748 loci. MLST typing was also performed, which is based on the nucleotide variation in seven conserved genes, the presence of plasmids (PlasmidFinder), and the presence of virulence genes (VirulenceFinder) on WGS data of each of the isolates [29]. The resultant cgMLST data files were organised in a matrix used as an input for visualisation of the loci variation and phylogeny construction based on the method outlined on the Norwegian Veterinary Institute (NVI) github page (https://norwegianveterinaryinstitute.github.io/BioinfTraining/R_trees.html, (accessed on 3 May 2021)) with slight modifications. Briefly, 1748 loci-based similarities were calculated from cgMLST files and a sequential comparison was performed to create a symmetrical dissimilarity matrix. Finally, hierarchical clustering based on the Unweighted Pair Group Method with Arithmetic (UPGMA) method was performed on all cgMLST data for all of the isolates. The isolate typing obtained by the cgMLST phylogeny was linked to clonal complexes assigned by cgMLST types and genetic lineages assigned by the Pasteur Institute classification scheme, available at: (https://bigsdb.pasteur.fr/cgi-bin/bigsdb/bigsdb.pl?db=pubmlst_Listeria_isolates&page=query, (accessed on 7 March 2023)). *L. monocytogenes* gene sequences previously shown to be linked to stress tolerance and virulence genes were collected from Listilist (http://genolist.pasteur.fr/, (accessed on 21 March 2023)). The presence or absence of virulence and biofilm formation genes [30] (see Appendix A) for each genome was established by performing a standalone local NCBI BLAST+ 2.10.0+ with a cut-off of >95% of nucleotide identity and an e-value of <10^−6^ (https://ftp.ncbi.nlm.nih.gov/blast/executables/blast+/LATEST/). Serogroups were assigned in silico based on the four marker genes, namely *lmo0737*, *lmo1118*, *ORF2819,* and *ORF2110* [31,32]. In addition, virulence, antibiotic-resistant, and Antimicrobial-Resistant (AMR) genes were identified by using a custom script to automatically loop the analyses of all genomes in the ABRicate program (v1.0.1) (https://github.com/tseemann/abricate, (accessed on 8 January 2023)) against the Virulence Factor Database (VFDB), Comprehensive Antibiotic Resistance Database (CARD), and antimicrobial-resistant (AMR) databases of NCBI [33,34]. 

To validate the presence of specific genes identified by the ABRicate tool that are unique to the CC-type in our isolates, fully assembled, high-quality complete *L. monocytogenes* genome sequences (*n* = 247) available at https://www.ncbi.nlm.nih.gov/data-hub/genome/?taxon=1639, (accessed on 16 January 2023) were acquired from the National Center for Biotechnology Information (NCBI) for similar analysis. Moreover, all available clinical isolates were specifically examined to link the presence of the genes (virulence, biofilm, stress tolerance, and antibiotic resistance) and evidence of human infections. The dataset of all *L. monocytogenes* (*n* = 98,297) from the Institut Pasteur MLST database accessible at http://bigsdb.pasteur.fr/Listeria (accessed on 9 February 2023) was exported for this purpose, and the clinical frequency was determined by dividing the clinically sourced isolates by the total of the clinical and food source isolates.

### 2.5. Statistical Analysis

Data visualisation and analysis were performed in R v.4.1.1. Fisher’s exact tests or Pearson’s chi-square were used for the association testing, and over-representation was identified by a Pearson residual value of more than 2. The alpha *p*-value threshold for significance was set at <0.05.

## 3. Results

### 3.1. Clonal Distribution and Phylogenetic Clustering

A 1748-gene-based cgMLST phylogenetic analysis on organic acid variability was reported previously with the same set of isolates used in this study [5]. This cgMLST phylogeny was replotted to visualise the Sequence Type (ST), clonal structure, and cgMLST alignment. Closely related CC-types, which were assigned by seven-gene in silico MLST and genetic lineages, were clustered and aligned consistently (Figure 1). To understand the genetic relatedness of CC-type to the presence of particular genes of interest, a pan-genome based phylogenetic tree was constructed using Roary analysis. As shown in Figure 2, the cgMLST tree generally showed a highly similar alignment to the Roary pan-genome phylogenetic tree. In particular, a comparison of CC-type distribution between the pan-genome analysis based on orthologous genes and a previously reported cgMLST-based phylogeny showed that CC-types from lineage I and lineage II are arranged in a similar way in both trees, whereas strains from lineage II showed more variability. In many instances, discrepancies were related simply to tree presentation and rotating clades could eliminate discrepancies across trees without changing their main structure (Figure 2).

### 3.2. Pan-Genome Analysis

Scoary (v1.6.16) results from a pan-genome-wide association study revealed 11 genes (Table 1) that were specifically associated (*p* < 0.05) with clinical isolates. Some of the Scoary-identified clinical-isolate-specific genes are similar to known virulence genes required for successful infection of human hosts but others of unknown function were also identified by the analysis. 

### 3.3. Presence of Virulence Genes and Their Genotype Association

In order to investigate potential associations of virulence genes with genetic grouping (CC, ST, Serogroup, Lineage), virulence genes were initially predicted in silico using VirulenceFinder. Eighty-nine virulence genes were found in the CC20, CC18, CC26, CC9, CC7, and CC155 isolates, as well as in the EGDe (CC9) reference genome. The fewest number of virulence genes were predicted in CC6 (seventy-seven). The number of virulence genes present in genomes of ST20 and ST18 isolates and reference genome EGD-e were the same as reported previously [38]. An independent *t*-test revealed that there are no significance differences in the presence of the total number of virulence genes in clinical versus non-clinical isolates. To fully understand the observed virulence gene differences, further analysis was performed with the ABRicate tool, which relies on the more limited, highly curated VFDB database. Only forty-one different virulence genes were predicted to be present in our isolates, including *Listeria* Pathogenicity Island LIPI-1, LIPI-2, and LIPI-3. Twenty-five of the forty-one virulence genes identified by ABRicate were found in all of the isolates. Interestingly, apart from the presence of twenty-five virulence genes in all the isolates, the remaining presence or absence was found to be CC-type-specific with very few exceptions (Figure 3 and Figure 4). In contrast, the presence and absence of the autolysin amidase gene (*ami*), collagen binding MSCRAMM (microbial surface components recognising adhesive matrix molecules) (*ecbA*), internalin family of surface proteins (*inlF*), one of the LIPI-3 genes *llsY*, and genes encoding the LPXTG surface protein (*vip*) were found to be ambiguous or not CC-type-specific. However, Fisher’s exact test revealed a significant association between the presence of *ami* with CC101, CC20, and CC7; *ecbA* with CC2, *inlJ* with CC101, CC18, CC54, and Novel CC-type; as well as LIPI-3 with CC1, CC3, CC4, and CC54. To corroborate such associations of genes specific to CC-types, a total of 247 assembled, high-quality, complete *L. monocytogenes* genomes were downloaded from the NCBI dataset and the same analysis was performed using the ABRicate tool. MLST typing using tseemann/mlst (available at: https://github.com/tseemann/mlst (accessed on 25 January 2023)) and the CC-type assignment based on the Pasteur Institute classification scheme revealed seventy different MLST types and forty-two different CC-types with five unassignable ST-types in the fully assembled NCBI genomes. The NCBI collection of strains shared the same number of virulence genes (*n* = 41), and the presence/absence of CC-type-specific genes was consistent across the strains used in this study (Appendix A). 

### 3.4. Antimicrobial Resistance, Antibiotic Resistance, and Biofilm Formation Genes

AMRFinderPlus and CARD databases in the ABRicate tool revealed the lincomycin resistance ABC-F-type ribosomal protection gene (*lin*) [39], fosfomycin resistance thiol transferase (*FosX*) [40], an integral membrane gene (*mprF*) [41], fluoroquinolones, and other structurally unrelated tetracycline resistant genes (*norB*) [39] in our isolates (Figure 4). A recent study reported that *L. monocytogenes* EGDe *lmo0919* is associated with resistance to lincomycin [42]. A standalone BLAST search in our isolates database revealed that *lmo0919* was exclusively present in the Lineage II isolates. It is worth noting that these resistance genes were found to be specific to cgMLST lineages (i.e., *lmo0919*, *lin*, and *norB* are present solely in Lineage II as well as *FoxS* solely in Lineage I). Specificity to lineage was also consistent among the NCBI *L. monocytogenes* genomes studied. BLAST searches (95% identity) for biofilm formation genes (Appendix A) revealed that all of the isolates carried a biofilm-formation-associated gene (*lmo0673*), the product of which catalyses the hydrolysis of S-ribosylhomocysteine to homocysteine (*luxS*), the cell-wall-binding gene (*lmo2504*), and the gene encoding a DNA repair protein (*recO*). The presence/absence of peptidoglycan binding genes *lmo0435* (*bapL*) and class 1 internalin (*inlL*) were found to be specific to CC-type; the presence of *bapL* was specific to CC121, CC14, CC204, CC9, and CC20. Likewise, *inlL* was exclusively present in the NOVELCC-type, CC155, CC26, CC37, CC18, CC204, CC20, CC412, and CC7 with the exception of reference strain 6179. The trend of genes specific to CC-type (*bapL* and *inlL*) was also found to be consistent when the same analysis was performed using NCBI strains (Appendix A). 

### 3.5. Stress Tolerance Genes

A genetic basis for acid tolerance and cold tolerance among the 150 *L. monocytogenes* isolates was investigated in our previous work [5]. The stress tolerance genes used in those studies are included in Appendix A. Apart from the subset of stress tolerance genes that are present in all the isolates, the stress tolerance genes were clearly specific to CC-type with the exception of an ATP-binding cassette (ABC) transporter and *gbuABC* operon [43] (Appendix A). In particular, *thiT*, *yycG*, *flhA*, *trpG*, *resE*, *betL*, *clpB*, *rpoN*_*sigL*, *trxB*, *OppA*, *itrABC*, and *deaD* were found to be specific to CC121, CC18, CC101, CC8, CC7, CC37, CC14, CC9, CC31, CC204, and CC20. It was noted that these CC types are phylogenetically close on both cgMLST- and Roary-gene-based phylogenies and are all Lineage II. 

### 3.6. CC-Types Associated with Human Listeriosis 

Adaptation to environmental stress enables *L. monocytogenes* to survive and even proliferate along the food chain. In particular, the capability for biofilm formation contributes to the persistence in food production facilities. Moreover, the presence of several virulence genes is important for the *L. monocytogenes* to survive within a host [44]. Since many of the analysed genes (virulence, biofilm, antimicrobial resistance, and stress tolerance) were found to be specific to clonal structure, we analysed all the *L. monocytogenes* data from the Institute Pasteur MLST database to calculate the frequency of human clinical isolation for each CC-type by dividing isolates from clinical sources by the total number of isolates. Analysis of 98,297 isolates showed that CC54 had the highest (100%) number of clinical isolates, followed by CC1 (76.94%), while the most prevalent clone, namely CC121 17.28%, had the lowest frequency (Figure 5).

## 4. Discussion

This study exploits WGS-based comparative genomic analysis of *L. monocytogenes* strains mainly isolated in the Republic of Ireland from food, food processing environments, and clinical sources and included CC-types known to associate with listeriosis outbreaks, namely CC1, CC2, CC4, and CC6 (Lineage I) and CC9 and CC121 (Lineage II), which are the most frequently isolated from immunocompromised patients [18,45]. It also included lineage II isolates that are prevalent in food processing environments belonging to CC9 [46,47,48], CC155 [49], and CC121 [18]. The analysis linked the presence or absence of key genes with specific CC-types and validated the linkage using genomes from the NCBI databases.

### 4.1. Heterogeneity in Virulence Genes

*L. monocytogenes*’ capability to colonise the gastrointestinal tract [50] and spread throughout the host dictates pathogen virulence. In the past couple of decades, multiple genes encoding virulence determinants have been described [19]. Most of the virulence genes are highly conserved but not all strains carry the full set. The important virulence factor LIPI contains several genes that encode virulence factors, including internalin proteins that allow the bacteria to invade host cells. For example, Listeriolysin O (LLO) specifically helps the bacteria escape from phagosomes and into the host cell cytoplasm, while other proteins enable the bacteria to replicate and spread within host tissues [51]. According to the ABRicate and CGE-based virulence-gene-finding analysis in this study, all of the virulence genes from LIPI-1 and LIPI-2 were present in all the isolates except *actA* from LIPI-1 and *inlJ* from LIPI-2. Both *actA* and *inlJ* encode essential virulence factors of *L. monocytogenes* [52,53] but were missing in all our CC1 and CC4 isolates, the most prevalent CC-type associated with human listeriosis globally [45,54], and this was consistent among the 247 NCBI genomes (Appendix A). However, standalone local BLAST with *actA* and *inlJ* genes from reference genome EGDe determined that these two genes were present in all genomes but with nucleotide identities as low as 95.18 and 86.18, respectively (Appendix A), which may be linked to the fact it has been reported that CC1, CC4, CC5, and CC59 strains possess disrupted versions of *actA* [55]. Another virulence gene found to associate with CC-type by ABRicate analysis was *inlF*, which was present in all CC-types except CC121, one CC3 isolate, and some CC59 isolates. Notably, among the isolates used in this work, CC121 and CC59, which are isolated from humans less frequently (2 out of 45), did not carry the *inlF* genes identified by ABRicate, which was in agreement with local BLAST searches (Appendix A). According to a recent study, *InlF* promotes entry into endothelial cells to breach the blood–brain barrier [56,57]. The *inlF* gene is part of a larger gene cluster, which includes the *inlA* and *inlB* genes. These three genes facilitate the entry of *L. monocytogenes* into host cells through interaction with distinct host surface receptors [53]. Since all of the isolates in this study carry *inlA* and *inlB* (Appendix A), the absence of *inlF* in the CC121 and CC59 group may limit successful infection. 

Overall, VirulenceFinder detected more genes than ABRicate, although none of the biosynthetic cluster involved in the production of Listeriolysin S (LLS) (*llsAGHXBYDP*) belonging to LIPI-3 [57] were predicted, because the VirulenceFinder analysis was performed (December 2022) utilising eighty-nine virulence genes based on the EGD-e reference genome, which does not encode LIPI-3. ABRicate predicted a total of forty-one virulence genes among the isolates of this study using the Virulence Factor Database (VFDB) including LIPI-3 genes exclusively in the Lineage I isolates CC3, CC54, CC4, CC6, and CC1. In fact, LIPI-3 is specifically present in most of the clinical isolates, where it is proposed to increase virulence [58]. However, it has also been found in food isolates [58,59] and CC-types with full LIPI-3, which may present the most significant risk. This study also found that isolates belonging to CC3, CC54, CC4, CC6, and CC1 were more likely to be isolated from cases of human listeriosis based on analysis of the Institute Pasteur MLST database (Figure 5). Notably, *inlF* and the full set of LIPI-3 genes are harboured by CC54, CC1, CC6, CC4, CC3, and CC224 (Figure 4), which showed a higher clinical isolation frequency of >46.66% (Figure 5) and are closely related by Roary-based phylogeny. However, other lineage II isolates such as CC90 and CC110 were also present at a higher clinical frequency in the Institute Pasteur MLST database even though these CC-types do not carry LIPI-3. It is possible that the genotype of lineage II CC-types harbouring more stress tolerance genes but fewer virulence genes than lineage I isolates (Figure 4) allows these organisms to persist in the food and food processing environments and they are still able to infect highly immunocompromised individuals [18]. 

### 4.2. Prevalence of Antimicrobial Resistance Genes

Antimicrobial resistance amongst *L. monocytogenes* strains appear to differ considerably [60] and multiple antibiotic resistance genes may contribute to the development of novel antibiotic-resistant strains [61]. Recent studies have reported the high prevalence of antibiotic resistance genes in *L. monocytogenes* [62,63]. The screening of AMR genes in the WGS by the ABRicate tool using CARD revealed that five different AMR genes were present in all the isolates. These AMR resistance genes may have evolved or transferred horizontally, resulting in an increase in antibiotic resistance [64,65]. The presence of other AMR genes in the WGS were generally found to be specific to genetic lineages. For instance, *lmo0919* [42] and *norB* [39] were harboured by Lineage II while *FosX* [63] was harboured by Lineage I. The link between WGS-based subtyping techniques and the presence of AMR genes may be important information for determining an appropriate treatment strategy for patients. 

### 4.3. Stress Response and Biofilm Formation Genes

When exposed to stress, *L. monocytogenes* develops strategies for altering cellular functions so that the pathogen can endure and proliferate [19]. It is able to adapt to stressful conditions, such as those encountered in food processing and storage environments, through the activation of specific stress response genes. In this study, all the Lineage II isolates, including CC20, CC18, CC26, CC9, CC7, and CC155, were predicted to contain more stress tolerance genes than Lineage I isolates (Figure 4). Unsurprisingly, these CC-type isolates have been observed to grow faster in cold temperatures [3] and demonstrate increased tolerance to acetic and propionic acid [5]. This may explain why these CC-types were more prevalent in food and food processing environments with the presence of combinations of more stress tolerance genes. In addition, *L. monocytogenes* is also capable of producing biofilm—extracellular polymeric substances (EPSs) [66] that encase the cells and shield them from environmental stresses such desiccation, pH fluctuations, and antimicrobial agents. Although the genes involved in *L. monocytogenes* biofilm formation are not yet fully characterised, a number of studies have identified an array of genes that could be involved (see Appendix A). Apart from the biofilm formation genes predicted as present in all the isolates, CC20, CC18, CC26, CC9, CC7, and CC155 types specifically possess class I internalin *inlL* biofilm formation genes potentially involved in mucin binding, sessile growth, and adhesion [64]. A similar linking of the presence of *inlL* to these CC-types was reported among *L. monocytogenes* in Norwegian food chains [65]. All of these CC-types belong to the most frequently reported serogroup 1/2a,3a in the European Union [66] and are second highest in the United States [67]. The combination of a higher proportion of stress tolerance genes and the presence of *inlL* in the genome may enable these CC-types to persist in the food processing environment and eventually infect immunocompromised individuals. 

## 5. Conclusions

Comparative genomic analysis of 150 *L. monocytogenes* isolates from food, the food-processing environment, and clinical sources was performed in this study. The relationship between CC-type and the prevalence of virulence, stress tolerance, biofilm formation, and antimicrobial-resistant genes was investigated. It was observed that several of these genes were clonal-structure-specific, and this specificity was verified using 247 completely assembled, high-quality genomes extracted from NCBI. Particularly, we highlight that CC20, CC18, CC26, CC9, CC7, and CC155 were predicted to harbour a higher proportion of biofilm, stress-survival, and antimicrobial genes than other CC-types. In addition, the CC-types, which carry both *inlF* and a full set of LIPI-3 genes, were found to have a higher human clinical isolation frequency than others. This WGS-based study provides a deeper understanding of this pathogen’s persistence within food chains and the ability to infect the host, but further in vitro adherence and invasion studies using host cell lines and in vivo studies using animal infection models are needed to support the in silico bioinformatics analysis.

## Figures and Tables

**Figure 1 microorganisms-11-01603-f001:**
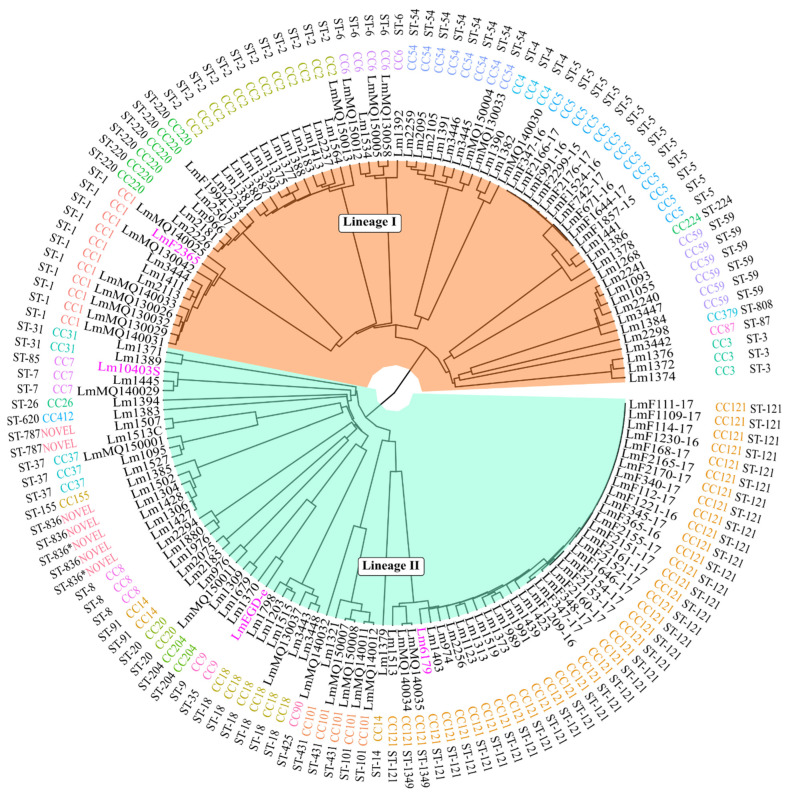
Unweighted Pair Group Method with Arithmetic Mean (UPGMA) tree based on the allelic profiles of the cgMLST target genes (*n* = 1748) of 150 *L. monocytogenes* isolates and their sequence type based on nucleotide differences in genomic sequences of seven different loci of housekeeping genes (MLST) along with assigned CC-type and genetic lineages. Reference strain names are highlighted in purple.

**Figure 2 microorganisms-11-01603-f002:**
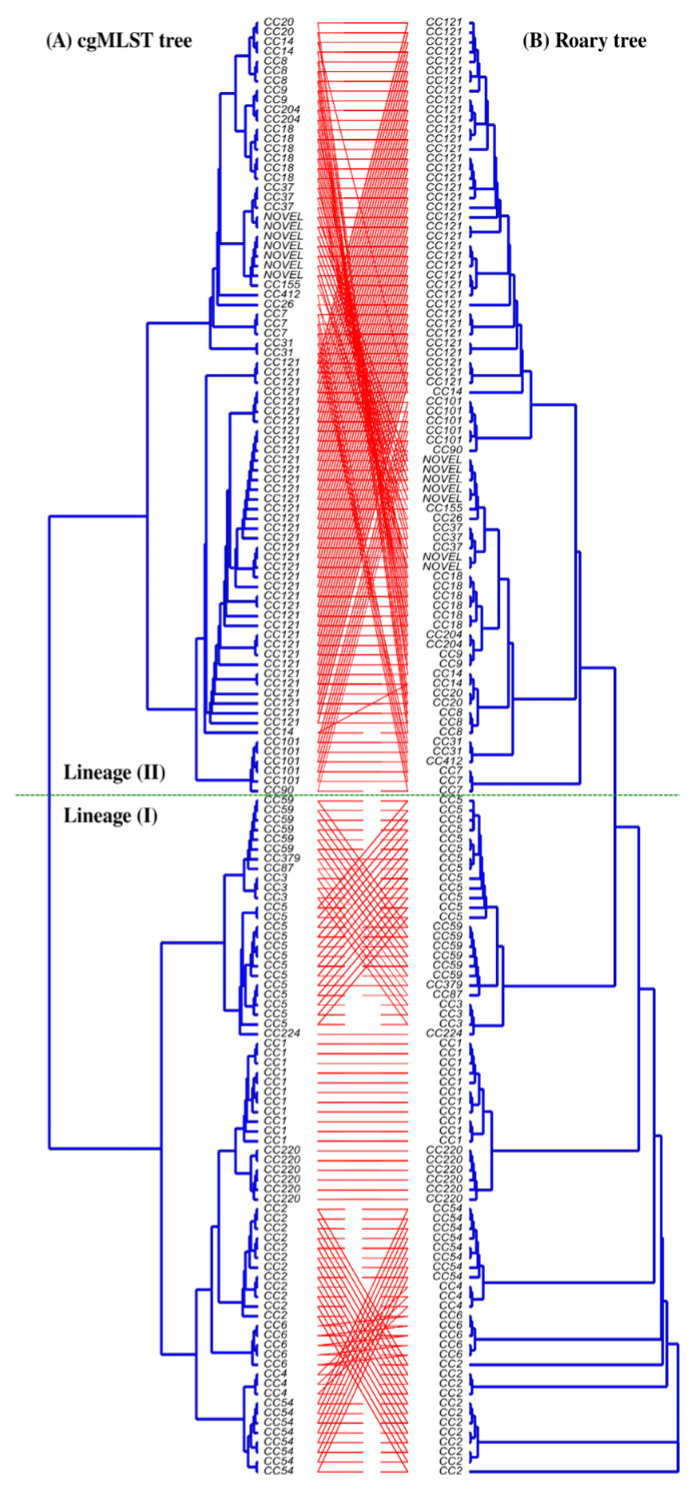
Comparison of phylogenetic trees constructed using the core genome Multi-Locus Sequence Typing (cgMLST) and Roary pan-genome-gene-based phylogeny for the 150 *L. monocytogenes* isolates. (**A**) Phylogenetic tree based on the UPGMA hierarchically clustered cgMLST of 1748 loci distances. (**B**) A pan-genome tree based on the presence or absence of genes in the Roary pan-genome pipeline.

**Figure 3 microorganisms-11-01603-f003:**
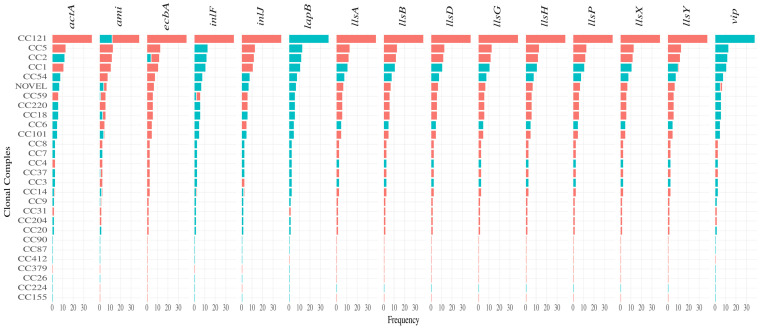
Variation in the presence (green) and absence (red) of virulence genes among CC-type in 150 *L. monocytogenes* isolates. The presence and absence of genes were determined by the ABRicate tool and using the Virulence Factor Database (VFDB) (http://www.mgc.ac.cn/VFs/, (accessed on 8 January 2023)).

**Figure 4 microorganisms-11-01603-f004:**
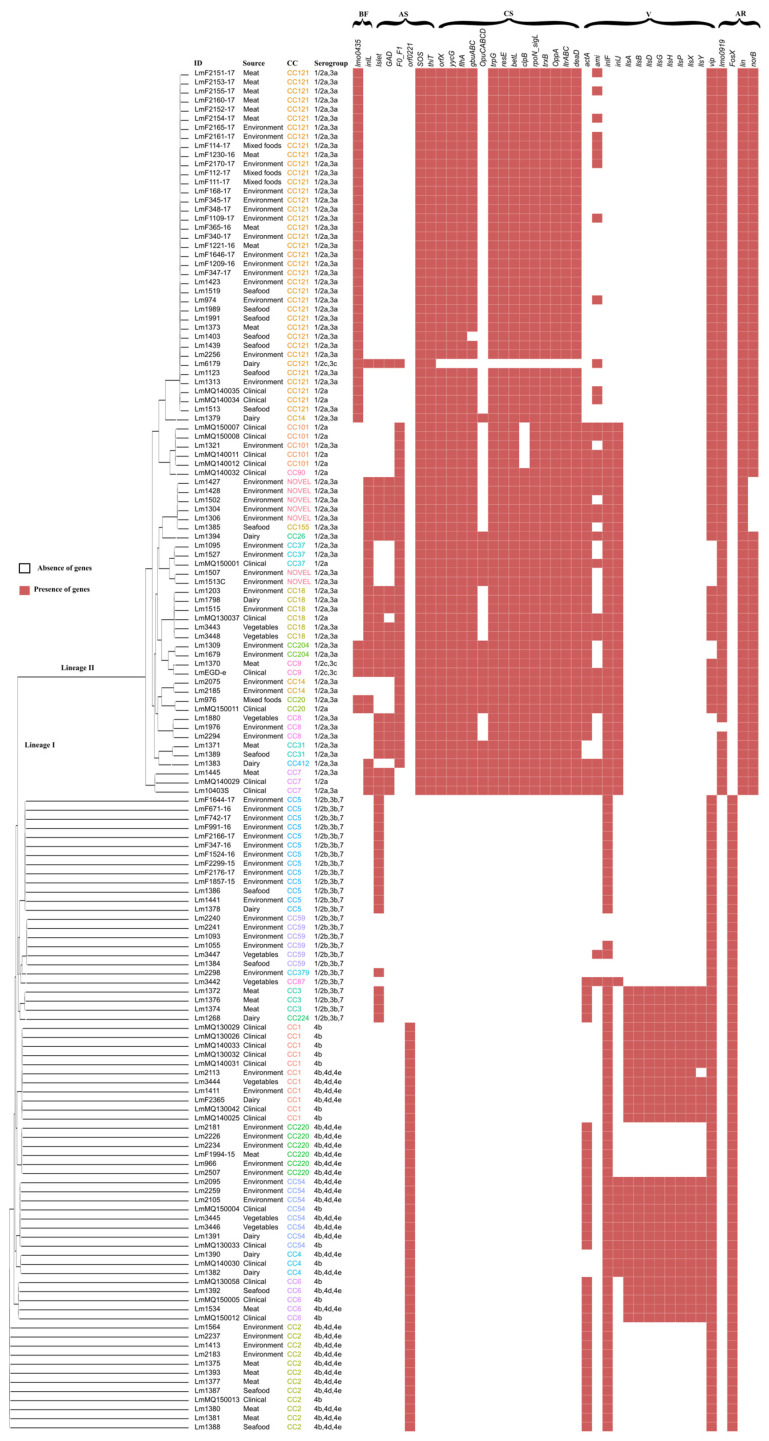
Phylogenetic relatedness and presence of respective genes among 150 *L. monocytogenes* isolates. Isolates ID, sources of isolation, MLST-based CC-type, and in silico-based serogrouping [31] are presented along with gene-based phylogeny constructed by Roary pangenome analysis. The presence of known biofilm (BF) formation, acid stress tolerance (AS), and cold stress tolerance (CS) genes identified by standalone BLAST and virulence (V) and antibiotic resistance (AR) genes identified by ABRicate (VFDB, CARD database) is indicated by red bars.

**Figure 5 microorganisms-11-01603-f005:**
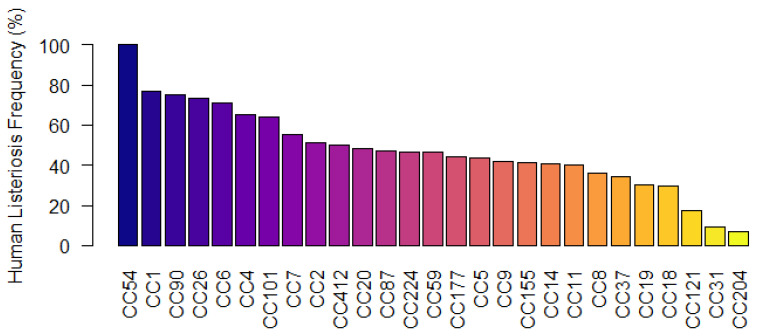
Human listeriosis frequency distribution analysis of *L. monocytogenes* (*n* = 97,397) based on CC-types (from PasteurMLST database available at https://bigsdb.pasteur.fr/cgi-bin/bigsdb/bigsdb.pl, (accessed on 9 February 2023)). Human listeriosis frequency for CC-types was calculated by dividing the number of clinical isolates by total number of isolates (food and human isolates).

**Table 1 microorganisms-11-01603-t001:** Pan-genome-wide association analysis identified genes specific to clinical isolates and their predicted function.

ID	Query (bp)	Accession	Identity (%)	Predicted Function
group_3	609	WP_242211654	100	LPXTG * cell-wall-anchored domain-containing protein
group_877	3276	WP_003726175.1	100	carbohydrate-binding protein
group_71	306	EEO3657903.1	100	transposase [*Listeria monocytogenes*] protein
group_1254	1515	WP_012951969.1	100	putative DNA-binding-domain-containing protein
group_10	2754	WP_014601151.1	99.89	autolysin Ami *
immR_1	423	WP_003724014.1	100	helix-turn-helix transcriptional regulator protein
group_902	210	WP_003733688.1	100	helix-turn-helix domain-containing protein
group_12	516	EAC8754143.1	100	GW-domain-containing glycosaminoglycan-binding protein
immA	426	EDD2318653.1	100	ImmA */IrrE family metallo-endopeptidase protein
group_3548	204	WP_222949761.1	100	hypothetical protein
ispDF	2592	EEW21888.1	100	LOW-QUALITY PROTEIN: peptidoglycan-bound protein

* Virulence contributor genes, *LPXTG* [35]; *Ami* [36]; *immA* [37].

## Data Availability

Genome sequence data was previously published [5].

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
