# Peer review of "Association of Virulence, Biofilm, and Antimicrobial Resistance Genes with Specific Clonal Complex Types of Listeria monocytogenes"

_microorganisms, 2023, doi:10.3390/microorganisms11061603_

Round 1
Reviewer 1 Report
General.
The manuscript is very long and thus difficult to read and tiresome. A lot of the information in there should be transferred to supplementary material, making the main text more reader-friendly.
The discussion should be divided into sub-sections to make it easier for reading.
As it is now, the manuscript cannot be considered for publication. It needs significant improvement and re-evaluation.
Moderate editing of English language.
Author Response
The manuscript is very long and thus difficult to read and tiresome. A lot of the information in there should be transferred to supplementary material, making the main text more reader-friendly.
Response: We appreciate that the scope of the manuscript is broad and addresses multiple diverse aspects of the Listeria monocytogenes genotype. However, the authors feel it was important to address the association between the strain genotypes and each of these key aspects (virulence, stress tolerance, biofilm formation and antimicrobial resistance) particularly given that the submission was targeting a special issue providing an update on L. monocytogenes. Every effort was made to ensure only essential information was included and in line with other reviewer suggestions, we have moved some of the material to supplementary material in the revised manuscript.
The discussion should be divided into sub-sections to make it easier for reading.
Response: We appreciate the reviewers suggestion for improving the manuscript and the Discussion has been reformatted into three sub-sections (4.1 Heterogeneity in virulence genes, 4.2 Prevalence of antimicrobial resistance genes and 4.3 Stress response and biofilm formation genes).
As it is now, the manuscript cannot be considered for publication. It needs significant improvement and re-evaluation.
Response: We have carefully revised the manuscript to improve clarity according to all the reviewer suggestions.
Reviewer 2 Report
The authors present an interesting study. In the future, such analysis will play major roles in the analytics of outbreaks. It would have been a great addition if some factors could have been strengthened with phenotypic data.
I suggest to accept the article after minor improvements (grammar and spelling errors).
Figure 1: Are the reference strains included as well? It would be good to mark them in the figure.
Table2: Too large for article. Should be supplementary.
Line 53-65: Would be great to see the “phenotypic” proof
L. monocytogenes can adapt to specific environments quite fast. This results in gaining and loosing functionality (or virulence). See https://www.frontiersin.org/article/10.3389/fmicb.2020.01726
Are there also indications on this in your tested isolates? It also seems that L. monocytogenes persisting already for a long time in dairy lose the ability to ferment pentoses. May there are also indications in your isolates?
The text is good to read and understandabel. However, there are several minor spelling and grammar errors (e. g. line 45: by mainly through consumption: it0s either by or through, not both).
Author Response
The authors present an interesting study. In the future, such analysis will play major roles in the analytics of outbreaks. It would have been a great addition if some factors could have been strengthened with phenotypic data.
Response: We appreciate the reviewers comment on the importance of these genetic association studies. The reviwer is also correct that phenotypic data relating to virulence and antimicrobial resistance would be very valuable. However, the primary focus of the paper was to look at associations between MLST type and presence or absence of specific genes. We would also highlight that significant stress tolerance phenotypic data is available (Myintzaw et al., 2022 and 2023, Wu et al., 2022) for these strains and a manuscript describing biofilm production is in preparation.
I suggest to accept the article after minor improvements (grammar and spelling errors).
Response: We have carefully reviewed the manuscript for grammar and spelling errors and made appropriate corrections.
Figure 1: Are the reference strains included as well? It would be good to mark them in the figure.
Response: We appreciate the reviewer highlighting this important point and have modified Figure 1 accordingly.
Table2: Too large for article. Should be supplementary.
Response: We have moved Table 2 to the supplementary material and updated the table numbers in the text.
Line 53-65: Would be great to see the “phenotypic” proof
Response: We agree with the reviewer that data relating to strain virulence would be very valuable but, as noted already, the primary focus of the paper was to look at associations between MLST type and presence or absence of specific key genes. In addition, the authors are mindful of ethical considerations in generating this data.
- monocytogenes can adapt to specific environments quite fast. This results in gaining and loosing functionality (or virulence). See https://www.frontiersin.org/article/10.3389/fmicb.2020.01726Are there also indications on this in your tested isolates? It also seems that L. monocytogenes persisting already for a long time in dairy lose the ability to ferment pentoses. May there are also indications in your isolates?
Response: The reviewer raises an interesting point and we have not looked specifically at loss of functions associated with strain origin. However, our results here and previous studies have failed to clearly identify specific metabolic genes or gene sets that are associated with source (Myintzaw et al., 2022)
Comments on the Quality of English LanguageThe text is good to read and understandable. However, there are several minor spelling and grammar errors (e. g. line 45: by mainly through consumption: it0s either by or through, not both).
Response: We have carefully reviewed the manuscript for grammar and spelling errors and made appropriate corrections.
Reviewer 3 Report
The authors investigate 150 Listeria monocytogenes isolates from various sources to understand their virulence, biofilm formation, and presence of antimicrobial resistance genes. They use Multi Locus Sequence Typing (MLST) to determine different clonal complexes (CC), identifying 28 CC-types, including 8 novel ones. The novel CC-types share similar stress tolerance genes and belong to genetic lineage II, serogroup 1/2a-3a. Analyses reveal 11 genes specifically associated with clinical isolates, and variations in the presence of Listeria Pathogenicity Islands (LIPI) and other virulence genes. Antimicrobial resistance gene analysis shows the presence of specific genes in different genetic lineages. Validation analysis with additional genome sequences confirms the consistency of findings.
In my opinion the article is well written. Furthermore, this study emphasizes the usefulness of MLST-based CC typing using Whole Genome Sequencing for classifying L. monocytogenes isolates. By combining MLST with WGS, authors obtain a detailed data connecting with genetic diversity, relatedness, and characteristics of bacterial isolates. I have only one issue for the authors: What message does this article convey? What are the further research plans based on the data obtained? Could the authors add a few sentences about this in the conclusions?
Author Response
In my opinion the article is well written. Furthermore, this study emphasizes the usefulness of MLST-based CC typing using Whole Genome Sequencing for classifying L. monocytogenes isolates. By combining MLST with WGS, authors obtain a detailed data connecting with genetic diversity, relatedness, and characteristics of bacterial isolates.
Response: We appreciate the reviewers comment on the importance of these genetic association studies.
I have only one issue for the authors: What message does this article convey? What are the further research plans based on the data obtained? Could the authors add a few sentences about this in the conclusions?
Response: We appreciate reviewer highlighting this important point. For clarity, we have added the following to the Conclusion “This WGS based study provides a deeper understanding of this pathogen's persistence within food chains and ability to infect the host but further in vitro adherence and invasion studies using host cell lines and in vivo studies using animal infection models are needed to support the in silico bioinformatics analysis.”
Round 2
Reviewer 1 Report
The manuscript only needs linguistic attention now before possible acceptance.
Moderate editing of English language required.
Author Response
I appreciate the reviewer's attention to detail very much. We are grateful. The authors have carefully reviewed the manuscript again and corrected a small number of grammatical errors in the revised manuscript. We believe the manuscript is now ready for publication.